# Evaluating Strategic Sampling of Conditional Diffusion Model-based Data Augmentation

**Aditya V. Kulkarni**[*1]

**Roger D. Soberanis-Mukul**[*1]          RSOBERA1@JHU.EDU

**Alex Pinsk**[2]

**Brittany-Lee Smith**[2]

**Ashlynn Cobb**[2]

**Alex Pacl**[2]

**Pranay Marlecha**[2]

**Jaiprakash Gurav**[2]

**Rashmi Sachan**[2]

**Therese Canares**[2]

**Mathias Unberath**[1]          UNBERATH@JHU.EDU

[1] *John Hopkins University, Baltimore, MD, USA.*

[2] *Curie Dx, Inc. Baltimore, MD, USA.*

**Editors:** Accepted for publication at MIDL 2025

## Abstract

The scarcity of large and well-annotated datasets is a concern in medical image analysis, particularly for emerging applications without substantial public dataset releases. Data synthesis has become relevant to address this problem, as conditional generative models can provide extensive amounts of data. However, the diversity of these synthetic samples can be limited to their training distribution, which restricts the benefits of synthetic data for augmentation. This paper analyses this limitation in the context of medical image classification using two datasets: chest X-ray and strep pharyngitis detection in smartphone photos. Our findings reveal that the performance improvements when augmenting training datasets with generated samples can be inconsistent. Furthermore, in some cases, using a small number of strategically chosen synthetic samples can outperform a larger, randomly selected synthetic sets. This highlights the need for effective sampling strategies in conditional diffusion models to improve training diversity and enhance performance in downstream applications.

**Keywords:** Diffusion Models, Diversity, Data Augmentation.

## 1. Introduction

The demand for large datasets in machine learning (ML) has motivated the creation of extensive open-source computer vision (CV) datasets. Pre-trained ML models can weakly annotate massive amounts of information, which humans can then validate (Kirillov et al., 2023). However, in the medical field, accurate annotation requires specialized knowledge, complicating a human-in-the-loop annotation process and motivating the need for data synthesis. While anatomical modeling and physics-based simulations can generate reliable

---

[*] Shared first author

synthetic data (Unberath et al., 2018; Killeen et al., 2023), they are not easily scalable. In contrast, statistical methods like conditional generative models (Rombach et al., 2021) enable scalable, potentially limitless labeled data generation and are well-suited for class-specific data augmentation on demand (Koetzier et al., 2024). However, the diversity—the ability of the model to create a variety of samples while still representing the actual data distribution (Ahmad et al., 2024)—is a concern because it may be limited to the variety of the factually observed samples during the generative model training. Recent works highlight a lack of diversity in samples generated using contemporary generative models, such as conditional diffusion models in real-world images (Geng et al., 2024; Sadat et al., 2024). This can reduce the effectiveness of data generation for downstream applications that utilize the generative model itself or the synthetic data it produces, like classification neural networks. Recognizing the relevance of diversity, the CV community has developed techniques to enhance sample variety, including guiding the diffusion process toward underrepresented areas and adding controlled noise during denoising (Qin et al., 2023; Sadat et al., 2024). However, while significant progress has been made in CV, the effectiveness of these strategies for improving downstream model performance through data augmentation, specifically in medical imaging, remains unclear. This paper evaluates diversity-driven data augmentation strategies for the Denoising Diffusion Probabilistic Model (DDPM) from the perspective of the downstream application. We compare three approaches: augmenting data with all samples generated by the DDPM, selectively filtering samples based on similarity to the training distribution, and filtering based on low-density regions. These techniques are implemented during DDPM inference, avoiding its retraining. We evaluated on the NIH Chest X-ray (Wang et al., 2017) (CXR) dataset and an in-house smartphone image dataset for strep pharyngitis (SP), obtained under approved protocols from the ethics committee of Johns Hopkins University (IRB00277755) and from industry partner CurieDx (22-CURI-101-CURI).

## 2. Methods

We use a DDPM based on Medical Diffusion (Khader et al., 2023). The model generates new samples in the latent space and incorporates a pre-trained Vector Quantized Generative Adversarial Network (Esser et al., 2021) (VQ-GAN) to encode and decode the image from this latent representation. We then evaluate three augmentation strategies. The first strategy directly samples the DDPM to randomly generate 10K samples for CXR and 3K samples for SP. The second strategy computes the closest distances based on the cosine similarity from the generated images to the DDPM training set and selects the synthetic data with the $k$-largest distances. The main motivation is that these images will provide complementary information to the DDPM training set. Finally, we evaluate a low-density-based selection strategy based on Density-Based Spatial Clustering of Applications with Noise (DBSCAN). DBSCAN generates clusters of low and high-density regions. We select the samples from the low-density—and potentially more diverse—regions to perform the augmentation. We evaluate the effects of these strategies on the medical image classification downstream application where an EfficientNet (Tan and Le, 2019) is trained to recognize five classes—Atelectasis, Cardiomegaly, Effusion, Consolidation, and Edema—for

Table 1: Baseline AUC and relative improvement (%) for the DDPM augmentation strategies. (*) indicates statistical significance under an AUC Delong test (DeLong et al., 1988) with corrections for multiple tests (Benjamini and Hochberg, 1995) with a $p$-value $< 0.05$. For the distance-based selection, we use the 10% and 20% most different samples.

| | Strep | Chest X-ray | | | | | |
|---|---|---|---|---|---|---|---|
| | Pos. | Atelect. | Cardiom. | Eff. | Consolid. | Edema | Avg. |
| Baseline AUC | 0.61 | 0.75 | 0.84 | 0.76 | 0.63 | 0.82 | 0.76 |
| **Strategy** | | Relative Improvement | | | | | |
| All Synth. | -3.3* | -0.72 | 2.68 | **1.35** | **5.17*** | -0.71 | **1.56** |
| 10% most diff. | **7.96*** | -0.49 | **2.90** | -0.83 | -0.25 | -0.47 | 0.17 |
| 20% most diff. | 1.6* | **0.73** | 1.47 | 0.01 | 0.25 | 0.58 | 0.32 |
| Low Density | 2.06* | -0.81 | 1.14 | 0.58 | 3.14* | **1.38*** | 1.09 |

CXR, and perform binary classification for SP. Details about the datasets are included in the appendix.

## 3. Results and Conclusion

We report the AUC for the baseline EfficientNet model trained on real data with standard augmentations, including random horizontal flips, 10-degree rotations, and Gaussian noise. For the DDPM augmentation strategies, we report the relative difference in AUC expressed as $100 * (AUC_{strategy} - AUC_{baseline})/AUC_{baseline}$. We used a fixed seed for all experiments to address training uncertainty in downstream classifiers, repeating the process with different fixed seeds three times for CXR and ten times for SP. Results are shown in Table 1. When evaluating the CXR dataset, we found that while some classes show statistical significance, the overall performance gains from synthetic data are modest. In contrast, the SP dataset, which is limited in size, shows significant improvements with selective augmentation strategies, whereas using all synthetic data leads to decreased performance. The three diversity-based approaches performed significantly better, with an 8% improvement when the 10% most diverse synthetic images were utilized. This suggests that most synthetic samples were redundant and added little value to the classification task. Given the dataset's size, naively adding generated images can lead to overfitting and worsen class imbalances. **Conclusion:** Synthetic data augmentation using generative models can help address the lack of labeled datasets in the medical field. However, recent studies suggest memorization and lack of diversity can limit effectiveness. Our analysis explores these effects in the medical field and shows that performance improvements can be inconsistent (as seen in the CXR dataset). While strategically selected augmentations can be beneficial in specific cases, like the SP dataset, these results emphasize the need for a better understanding of diversity and memorization in different tasks and highlight the need for effective sampling and selection strategies to improve generative data augmentation techniques.

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

## Appendix

### Datasets Specification

**NIH Chest X-Ray Dataset.**   We perform experiments on a subset of the NIH Chest X-ray Dataset for multi-class downstream tasks. We selected the Atelectasis, Cardiomegaly, Effusion, Consolidation, and Edema classes to train the models (DDPM conditioning and downstream models). The training set includes 8280 Atelectasis, 1707 Cardiomegaly, 8659 Effusion, 2852 Consolidation, and 1378 Edema samples. The test set has 3279, 1069, 4658, 1815, and 925 samples, respectively. Both DDPM and the classifiers were trained on the full training set, with the test set split into 25% validation and 75% test for the downstream classification task.

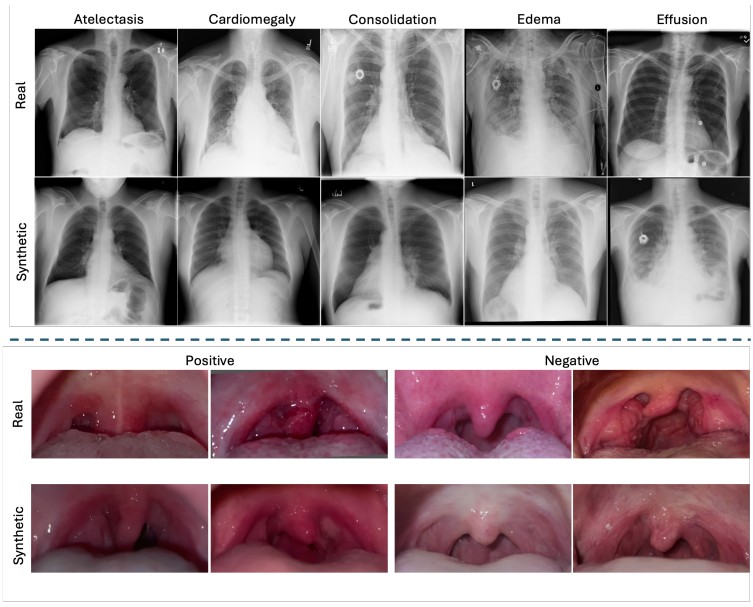

Figure 1: Examples of real (top) and synthetic (bottom) examples for the different classes. **Top**: Chest X-ray, **Bottom**: Strep pharyngitis.

**In-House Strep Pharyngitis.**   The in-house dataset is composed of images from 883 patients for the downstream task of strep pharyngitis classification. Streptococcus pharyngitis, or strep pharyngitis, is a bacterial infection of the throat that is confirmed with a rapid antigen test or throat culture. A total of 527 of these patients have labels for strep pharyngitis. Each image was captured with a cellphone, focusing on the throat area. For each image, we cropped the image to focus only in the tongue/tonsil region, and removed the images that were out of focus or lacked the region of interest. The data was collected at the Johns Hopkins Hospital under the approved protocol IRB00277755 and from CurieDx under IRB protocol 22-CURI-101-CURI. Informed consent was obtained from the participants before obtaining the images. We performed a random patient-wise training and testing partition. In total, the training set contains 150 strep positive patients (14533 images) and 213 strep

negative patients (22035 images), while the testing set includes 66 positive (6867 images) and 98 negative (10307 images). In each training iteration, we randomly select a single frame from the multiple images available per patient. However, training epochs remain organized on a patient-wise basis. The Figure 1 shows Examples for both datasets together with samples generated using the DDPM.

