# OpenReview forum: "Evaluating Strategic Sampling of Conditional Diffusion Model-based Data Augmentation"
_MIDL.io/2025/Short_Papers — MIDL 2025 - Short Papers_

### Official Review · Reviewer_6RPd · 2025-04-16

**Rating:** 4
**Confidence:** 4

**Summary:**

The paper evaluates different strategies for augmenting medical imaging datasets with synthetic data generated by conditional diffusion models. The authors focus specifically on addressing diversity issues in synthetic datasets to enhance downstream medical image classification tasks, using Chest X-ray and strep pharyngitis detection datasets.

**Strengths:**

The paper addresses relevant diversity and strategic sampling of synthetic data for improving downstream tasks, which is highly significant in medical imaging due to limited annotated datasets. It clearly demonstrates scenarios in which strategically selected synthetic samples outperform random synthetic augmentation. The experiments and results clearly show potential benefits, particularly for smaller datasets.

**Weaknesses:**

On the larger CXR dataset, gains are limited, highlighting the challenge of synthetic augmentation in well-represented domains.

Only two datasets are tested, both image-based, with no exploration of other modalities or tasks (e.g., segmentation, detection).

While the premise is about improving diversity, there’s no explicit metric for diversity or memorization—only performance proxies are used.

---

### Decision · Program_Chairs · 2025-05-01

Accept